# The Influence of Emissions from Maritime Transport on Air Quality in the Strait of Gibraltar (Spain)

Vanessa Durán-Grados [1], Rubén Rodríguez-Moreno [1], Fátima Calderay-Cayetano [1], Yolanda Amado-Sánchez [1], Emilio Pájaro-Velázquez [1], Rafael A. O. Nunes [2], Maria C. M. Alvim-Ferraz [2], Sofia I. V. Sousa [2] and Juan Moreno-Gutiérrez [1],*,†

[1] Departamento de Máquinas y Motores Térmicos, University of Cádiz, 11519 Cádiz, Spain
[2] LEPABE—Laboratório de Engenharia de Processos, Ambiente, Biotecnología e Energía, University of Porto, Rua Dr. Roberto Frias, 4200-465 Porto, Portugal
* Correspondence: juan.moreno@uca.es
† Current affiliation: Thermal Engines Department Polígono Río San Pedr, University of Cádiz, 11510 Puerto Real, Spain.

**Abstract:** Gaseous and particulate emissions from oceangoing ships have a significant effect on the quality of air in cities. This study estimates mainly the influence of $NO_x$, $SO_x$, and particulate matter ($PM_{2.5}$) on air quality in the Strait of Gibraltar (Spain) using the authors' own Ship's Energy and Emissions Model (SENEM) and the California Puff air quality model (CALPUFF) in 2017. The Algeciras Bay Industrial Zone recorded the highest levels of pollutants, and the Palmones area was identified as a major hotspot, with mean daily ship-sourced $SO_x$ concentrations >215 µg/m$^3$, while the highest concentrations of $PM_{10}$ reached 8.5 µg/m$^3$ inside the Strait, and the mean yearly contribution of $PM_{2.5}$ reached 0.86 µg/m$^3$ in the city of Algeciras. The incidence of maritime traffic emissions on the levels of particle emissions, CO, HC, NMVOC, and $CO_2$ reached values of up to 20–25% in all the receivers defined in the study.

**Keywords:** shipping emissions; air quality calculation model; GHG; emissions calculation model; pollutants from ships



## 1. Background and Introduction

Ships' propulsion systems have a negative impact on air pollution [1,2] and GHG emissions [3] and long-term effects on human health and environmental quality. Overall deaths due to these effects are estimated to have risen to 87,000 deaths per year from 2007 to 2013 [4].

Air pollution, both indoors and outdoors, is the largest environmental risk to public health (diseases and conditions that make us physically unwell, such as infections, allergies, tiredness, headaches, respiratory issues [5], etc.). It produces both short-term and long-term illness and potentially reduces life expectancy. Air quality is determined by the environmental conditions and the amount of particles and polluting gases that the air contains. The maritime transport sector, which is one of the least-regulated anthropogenic sources of emissions, contributes significantly to air pollution, particularly in coastal areas near major ports. In this sector, a recent study [6] estimated that maritime transport emissions accounted for 3500 premature deaths from $PM_{2.5}$ and $O_3$ across the USA in 2013. This study is a contribution to any current or future epidemiological analyses carried out in the area of the Strait of Gibraltar, which can be considered an area of major scientific interest. Epidemiological studies are based both on the results of emission models in which pollutants (from ships in this case) are quantified one by one and serve as input data for the air quality model. The extensive epidemiology literature has also documented the association of fine particulate air pollution with mortality. Most of this research consists of time series studies of the effects of particle exposures experienced over a relatively few

days before death. Three follow-up cohort studies in the United States and a recent pilot study from Europe evaluated the effects of long-term average ambient concentrations of fine particles and other air pollutants over many years [7].

On the other hand, European maritime emissions are a significant share of global ship emissions of GHGs; shipping emissions contribute to local air quality problems; and at a global scale, ship emissions of both air pollutants and GHGs have a net cooling effect [8].

Therefore, the promotion of sustainable shipping and sustainable maritime development is one of the major priorities of the IMO in the coming years [9]. The reason for this is that the future growth of ship traffic will affect the composition of the atmosphere, contributing to worsening air quality. These emissions are also generated while vessels are at berth and affect not only major ports, but also medium- and small-scale ports [10]. Approximately 13%, 12%, and about 3–4% of total global emissions of $NO_x$, $SO_x$ [11], and PM [12], respectively, are emitted by ships. Most (87%) of these are attributed to international shipping activity [13], while domestic shipping was responsible for about 9% of total shipping emissions of $CO_2$ and fishing for around 4%. Examining the makeup of the shipping fleet reveals that 55% was generated by container ships, bulk carriers, and oil tankers. Note that the available time data do not have discounted periods of Saharan intrusion dust. However, very little is known about the magnitude and effects of air pollution due to marine vessels [14].

Comparing European Union (EU) standards and the new World Health Organization (WHO) air quality guidelines published in 2021, the former are less strict than the latter. In 2019, 97% of the EU urban population was exposed to concentrations of fine particulate matter ($PM_{2.5}$) above the new WHO guideline level of 5 $\mu g/m^3$ [15].

On this topic, it is also relevant to take into account the results of a study applied for the Iberian Peninsula [16] showing that in the case of the Strait of Gibraltar, $CO_2$, $NO_x$, sulfate, and $SO_x$ had the highest values: 1330, 24, 1.03, and 11.6 t $yr^{-1}$ $km^{-2}$, respectively. For these reasons, the authors decided to use two different models from those of said study and exclusively applied them to the Strait of Gibraltar area.

Since the world-famous Strait of Gibraltar has a long maritime history and currently plays a crucial role in shipping trade, we have considered it necessary to conduct the air quality study presented here. The Strait is one of the main shipping traffic lanes in Europe (115,708 ships/year; 2017; www.gibraltarport.com). In this case, only 92,000 ships were analysed. This study provides new information (air quality) on this traditional water lane (no war ships, fishing vessels, dredgers, tugs, or auxiliary boats).

An epidemiological study from our own published research paper [17], applied to the Iberian Peninsula, concluded that in terms of premature mortality per 100,000 inhabitants, $NO_2$ air pollution contributed to 36.5 deaths, 48.8 deaths, and 57.5 deaths in Barcelona, Valencia, and the Strait of Gibraltar, respectively. For all-cause mortality, $PM_{2.5}$ emissions from ships contributed to 12.5, 20.4, and 24.1 deaths per 100,000 inhabitants in Barcelona, Valencia, and the Strait of Gibraltar, respectively.

In this study, the resulting concentrations will be applied to the epidemiological model. There are 16 air quality monitoring stations in the area studied, whose accuracy of measurements will largely depend on weather conditions. However, this study quantifies emissions by emission focus (ship) first. These results are introduced as input data in the air quality model in such a way that the resulting measures of the stations can differentiate those that come from the marine transport and those that come from other industries. This is the most significant contribution of this study. In the section "Results and Discussion", the different situations are analysed. To develop this study, it is necessary to use two models: (a) an emission model and (b) an air quality model. SENEM's own model for calculating emissions and CALPUFF (both are described in Supplementary Material) for air quality were used in this study.

In order to obtain results that are as true to life as possible, this study uses the SENEM model to quantify emissions from ships. This model, unlike other models, such as STEEM [18] or STEAM [13], quantifies the power delivered by propulsion engines taking

into account all the parameters which influence the resistance to the ship advancing. The complete procedures are defined in the SENEM model published by the authors of this paper [19].

In the case studied in this manuscript, based on the SENEM, such assessments are based on air quality dispersion models in which the amounts of primary pollutants ($CO_2$, $CH_4$, $N_2O$, NMVOC, $NO_x$, $SO_x$, CO, and PM) emitted directly into the atmosphere are calculated using a bottom-up approach (inventories compiled from ship activity records and activity-based emission factors for different ship types); these data serve as the main input for the models [20].

The main objective of this study is to quantify the influence of emissions from ships on air quality for each emitted pollutant across the Strait of Gibraltar, the limits of which are shown in Figure 1. In order to describe the maritime traffic in the Strait of Gibraltar in detail, the first step was to develop an inventory of shipping routes and their characteristics observed ship-by-ship. Databases from the Automatic Identification System (AIS) crossed with the Lloyd's Register Fairplay allowed for the highly accurate reconstruction of almost 100% of the routes of high-tonnage vessels cruising in the Strait of Gibraltar during 2017.

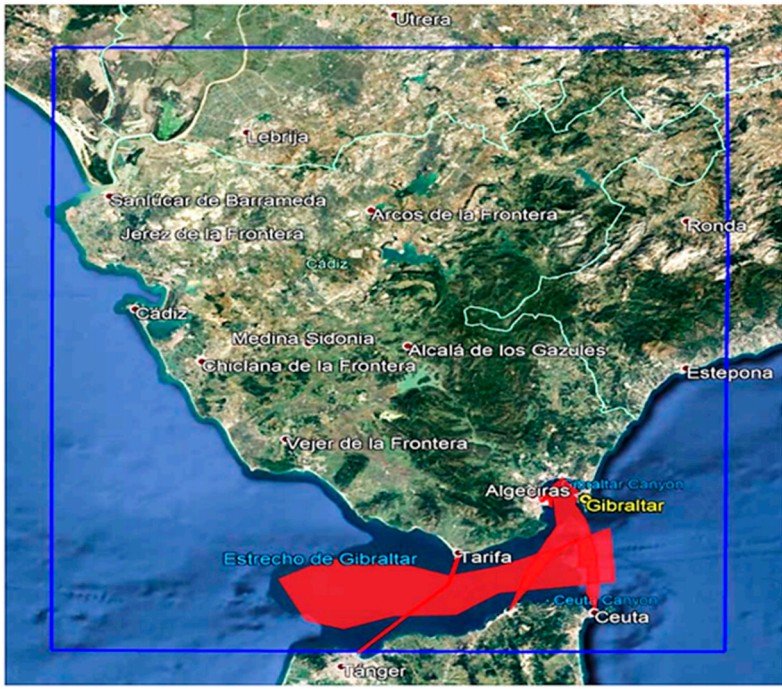

**Figure 1.** Strait of Gibraltar–area studied.

To quantify the concentration of each pollutant in the air, the second step involved the use of the CALPUFF chemistry–transport model. CALPUFF simulated all the scenarios envisaged in the Strait of Gibraltar.

## 2. Methodology

Inventories of shipping emissions were compiled using shipping traffic data from 2017 obtained from the Automatic Identification System (AIS); 92,000 individual ship movements were recorded. In this case, primary pollutants ($CO_2$, $CH_4$, $N_2O$, NMVOC, $NO_x$, $SO_x$, CO, and PM) were studied.

The Strait of Gibraltar is a narrow stretch of sea that links the Atlantic Ocean to the Mediterranean Sea and is bordered by the Iberian Peninsula to the north and Morocco to the south (Figure 1). It covers an extension of 2640 km$^2$ and is about 60 kilometres long and between 14 and 44 kilometres wide. The total population in the region in 2017 was about 2,349,738 inhabitants, distributed across 4 provinces: Ceuta, 77,389; Gibraltar, 32,194; Tangier, 1,000,000; and Cádiz, 1,240,155.

### 2.1. Emissions Model

To study the contribution of maritime and port traffic in the area of the Bay of Algeciras and the Strait of Gibraltar on air quality in the province of Cádiz, it is necessary to first identify the emission sources to consider and quantify their corresponding emissions. To minimise uncertainties, producing an emissions inventory that is as accurate as possible is very important. For these reason, it is necessary to know the total amount of emissions attributable to shipping activities [21].

Four main shipping routes have been identified that could affect pollutant levels in the province of Cádiz (Figure 1):

- Strait of Gibraltar: ships crossing the Strait in an east–west–east direction (without docking or heading for port).
- Algeciras Port: passenger ships, container ships, and tankers (manoeuvring and berthing in port).
- Tarifa–Tangier shipping lane: boats sailing the Strait in a north–south–north direction, covering the Tarifa–Tangier route.
- Ceuta Port: boats crossing the strait in a north–south–north direction, covering the Ceuta Port route.

Therefore, for the definition of the scenario to be simulated, emissions are considered under normal conditions for the months of March to October and under adverse weather conditions for the months of January, February, November, and December.

This study uses the SENEM model [19], using g/kWh as the base unit of emission factors. SENEM takes into consideration a range of factors affecting the value of the main engine load factor, such as hull and propeller maintenance and the efficiency of the propulsion system, waves, current, and the wind. The SENEM model calculates the emission from ships in all modes of navigation: cruising, manoeuvring, anchoring, and at berth.

This model was validated and compared with the Ship Traffic Energy and Environmental Model, STEEM [18], and the Ship Traffic Emissions Assessment Model, STEAM [13], in a study applied to domestic traffic conducted in the Strait of Gibraltar [19].

In this sense, the ships at berth also emitted pollutants due to hoteling activities. The high concentration of ships in port areas means that they receive a significant amount of these emissions. This is the case of Algeciras, the most important port of the Strait of Gibraltar.

Emissions during 2017 were modelled using data of the pollutants emitted calculated by the SENEM model. This model quantifies the power delivered by propulsion engines, taking into account, among other factors, the weather conditions (Equation (S1), Supplementary Material).

During most months of the year, the weather conditions did not affect the energy consumption, but for 120 days mainly in January, February, November, and December, the weather conditions were very strong, as the wind rose shows (Figure 2). The speed loss was of one knot, and for this reason, to maintain the same speed, the power delivered increased, as did fuel consumption and emissions.

As an example, a comparative study developed [22] on six ships in the Strait of Gibraltar showed the total energy consumption (kWh) both for calm water and taking into account the worst weather, hull, and propulsion system performance conditions. Differences of up to 20% were found in some cases.

In this case, by applying the SENEM model, a one knot loss in speed for each ship was considered when Equation (S1) [19] was applied.

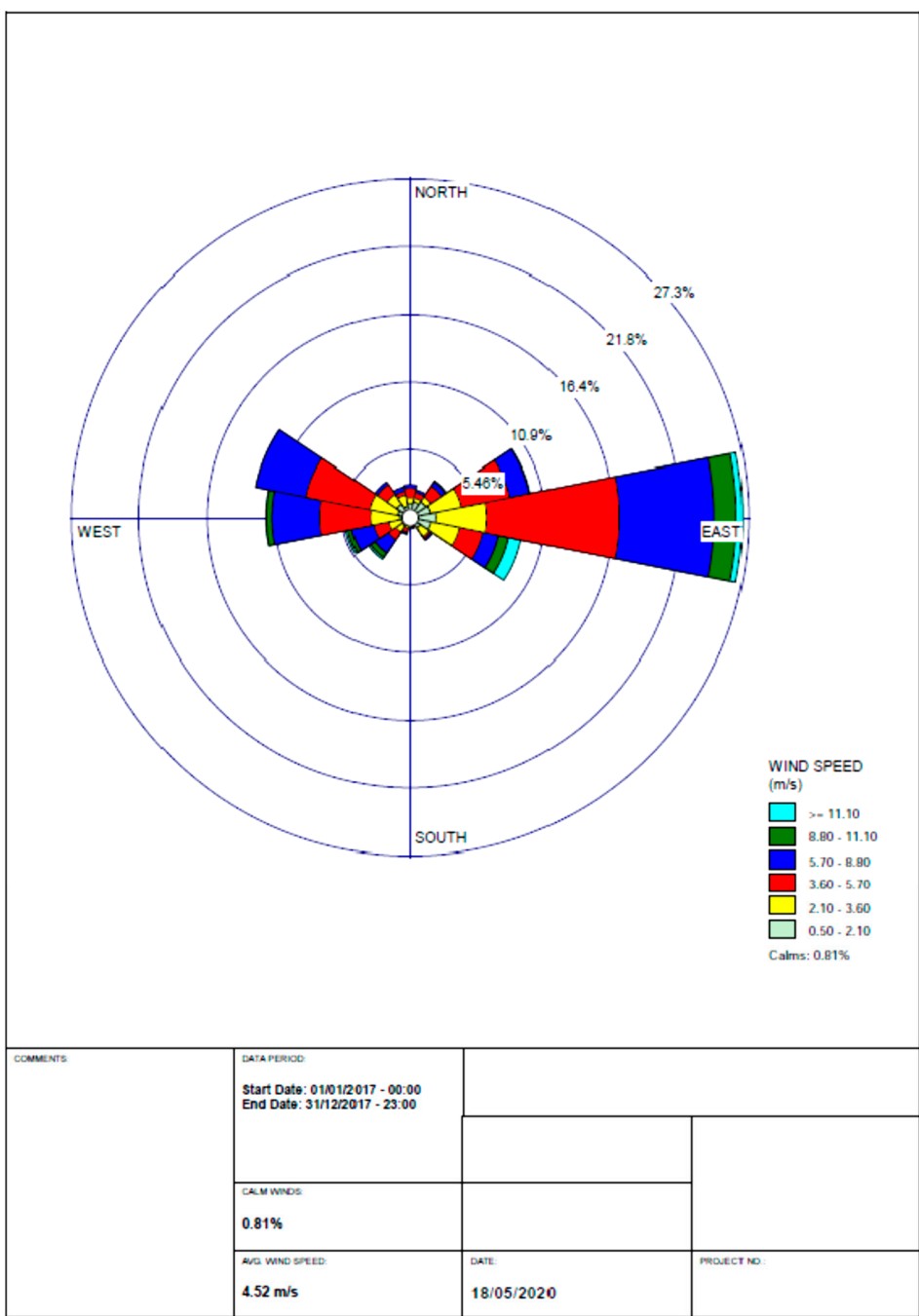

**Figure 2.** Wind rose in the Strait of Gibraltar during 2017.

### 2.2. Air Quality Model

To calculate the effects of emissions from these ships, dispersion models can be used. For example, ISCT3 and AERMOD v.5.2 [23], SMOKE [24], CMAQ-DDM, and SILAM, [25]. The CALPUFF model was used in this case (Figure 3). The methodology followed to perform the study is described in the Supplementary Materials.

The physical and chemical processes are simulated using CALPUFF, which uses mathematical and numerical techniques to simulate the dispersion of air pollutants into the atmosphere and how they react. They are based on meteorological inputs and source information from ships' emissions. However, there is considerable uncertainty surrounding both the modelling of air quality and the compilation of inventories of shipping emissions.

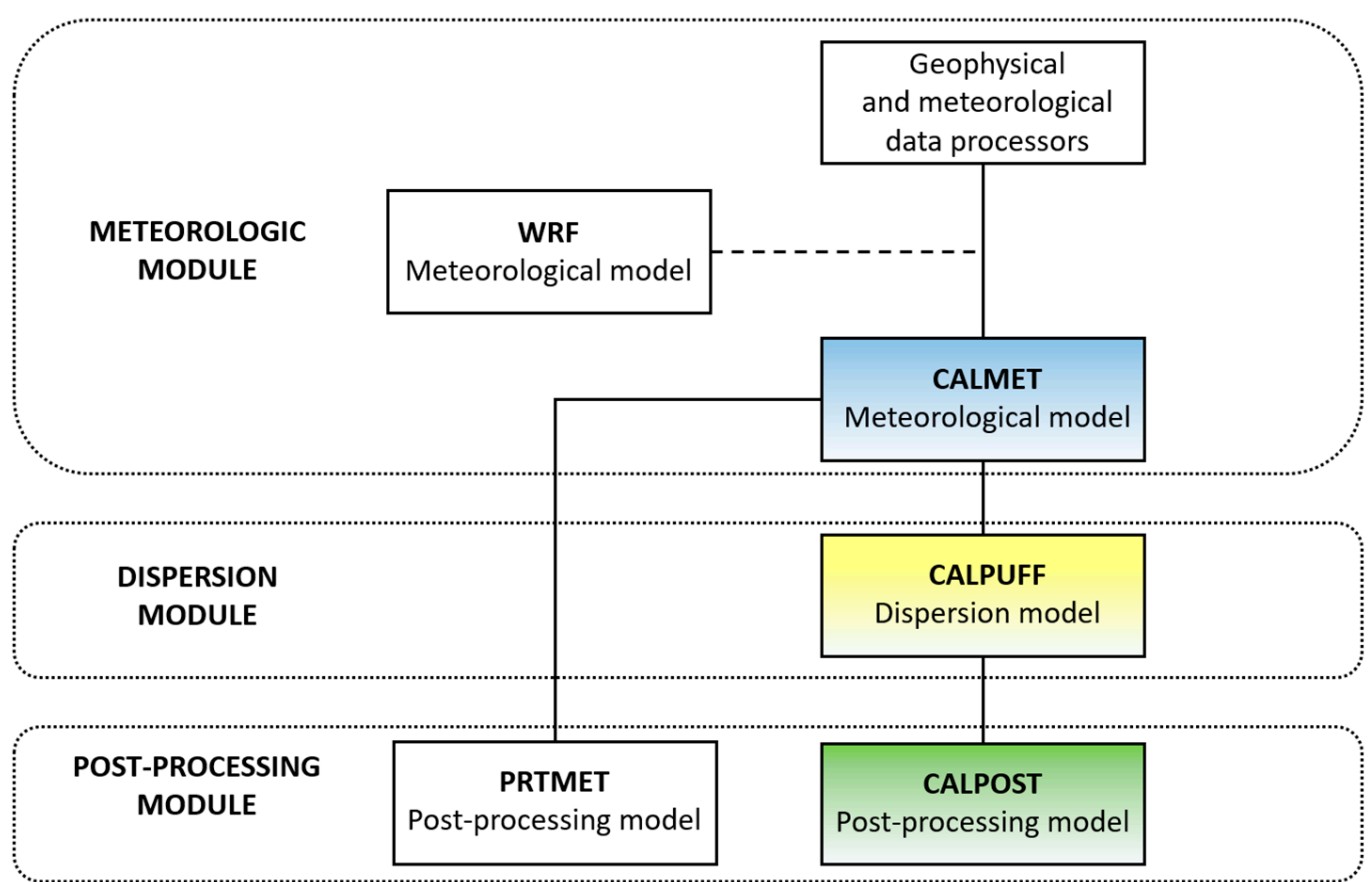

**Figure 3.** CALPUFF air quality model.

With regards to the epidemiology model, a pilot study from Europe involving a long follow-up period reported an increase in mortality among subjects subjected to high ambient concentrations of fine particles and other air pollutants over many years [7].

The dispersion of ship emissions was modelled using CALPUFF [26], one of the preferred models of the U.S. Environmental Protection Agency (U.S. EPA) [27].

The model was designed for two scenarios: considering shipping emissions and not considering shipping emissions and had a horizontal resolution of 500 m × 500 m (long–lat) and an hourly data output for 2017. Emissions from other sources were obtained from the Andalusian Air Quality Monitoring and Control Network Stations (RVCCAA) [28].

To design an air quality model to simulate how seafaring traffic contributes to the levels of pollutants emitted, the following steps were followed: (a) selection of the studied area, (b) characterization of the meteorological conditions in these area, (c) selection of the topography and use of the land, (d) characterization of the sources of emissions, and (e) definition of the interest points (discrete receptors, Figure 4).

The meteorology in the Strait of Gibraltar area plays a fundamental role, particularly in the dispersion of pollutants. Thus, to achieve a more accurate measurement, a dispersion model together with a meteorological module (CALMET) was used. The methodology used is described in the Supplementary Material.

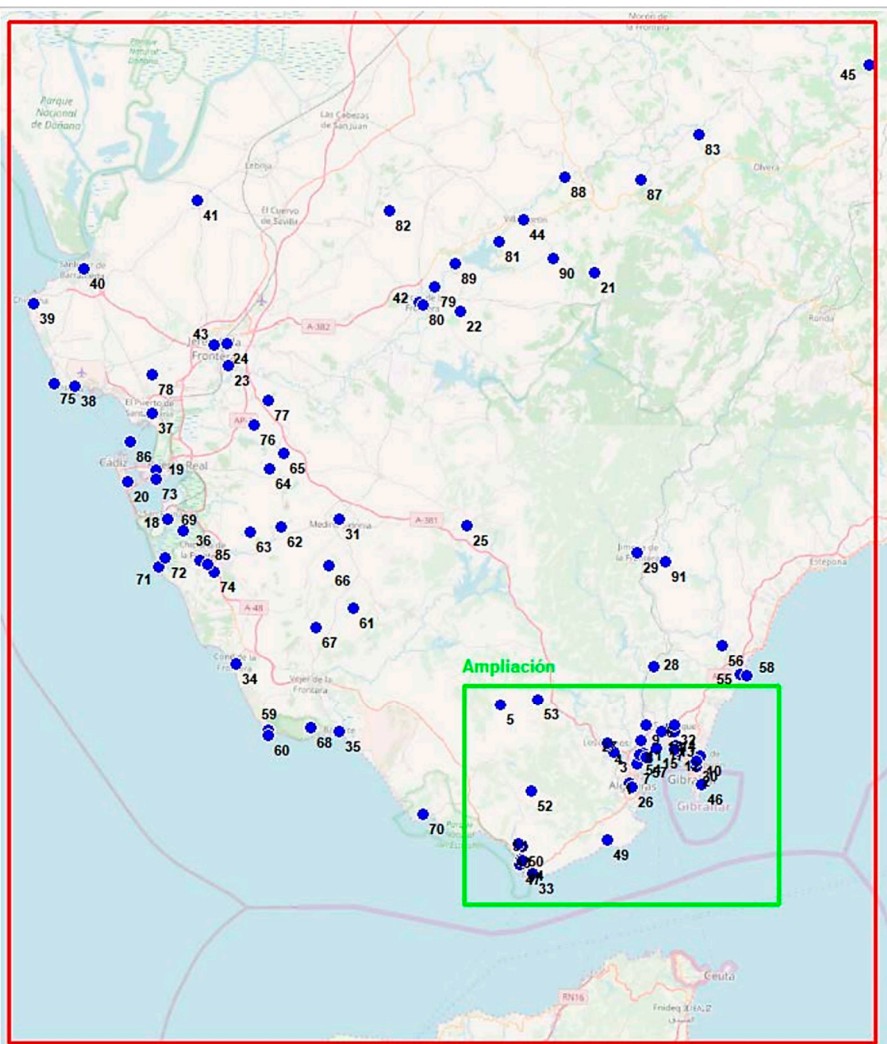

**Figure 4.** Discrete receptor locations.

## 3. Results and Discussion

### 3.1. Quantifying Pollutant Emissions

To avoid confusion or redundancy between the models used (SENEM and CALPUFF), it is convenient to consider that the SENEM model data are only used as input data for the CALPUFF model.

Once the CALPUFF dispersion model was calibrated, the contribution of pollutant emissions from maritime traffic to air quality levels in the study area was measured. The pollutants analysed in the study, based on the fuels usually used by ships, are $SO_x$, $NO_x$, CO, $PM_{10}$, $PM_{2.5}$, HC, NMVOC, and $CO_2$. In order to show the increase in these pollutants, the CALPUFF model was configured both considering and not considering shipping emissions.

It would be appropriate to note that the highest contributions of shipping emissions to the concentration levels were recorded during spring and summer (18% higher than during winter, Table 1). This pattern seems to be related to the increase in ship traffic during the summer, which led to increased contribution to the pollutant concentration levels.

**Table 1.** Seasonal amounts of emitted air pollutants from shipping in the Strait of Gibraltar, including ships at berth, in 2017 (t yr$^{-1}$).

| Pollutant. | Spring | Summer | Autumn | Winter | Total |
|:---:|:---:|:---:|:---:|:---:|:---:|
| $NO_x$ | 4150.2 | 4686.1 | 4058.1 | 3866.4 | 16,760.8 |
| $SO_x$ | 1195.5 | 1350 | 1169 | 1113.9 | 4828.4 |
| $PM_{10}$ | 144.6 | 163.25 | 141.36 | 134.69 | 583.9 |
| $PM_{2.5}$ | 23.15 | 26.14 | 22.63 | 21.58 | 93.5 |
| $CO_2$ | 207,643.6 | 234,462.4 | 202,979.5 | 193,561.6 | 838,647.1 |
| CO | 184.26 | 208 | 180.1 | 171.84 | 744.2 |
| HC | 167 | 191.9 | 166.1 | 161.6 | 686.6 |
| NMVOC | 146.9 | 168.8 | 146.1 | 142.2 | 604 |

For calculating auxiliary engine power, Table S6 (Supplementary Material) was used.

The SENEM model used the Kwon method [29] to predict speed loss due to added resistance in abnormal weather conditions (irregular waves and wind) when Equation (S1) [19] was applied. The Kwon model has the advantage that it is easy and practical to use.

The total results for the Strait of Gibraltar can be found in Table 2. This table shows the emissions of $PM_{2.5}$, $SO_x$, and $NO_x$ for eight months of calm water conditions and four months in which the weather was bad. In 2017, 590 t/year of $PM_{2.5}$, 4830 t/year of $SO_x$, and 16,760.8 t/year of $NO_x$ were emitted in the Strait of Gibraltar.

**Table 2.** Pollutant emissions from maritime transport in the Strait of Gibraltar in 2017 (kt).

| Course | $CO_2$ | $NO_x$ | CO | HC | $PM_{2.5}$ | $SO_x$ | NMVOC |
|:---:|:---:|:---:|:---:|:---:|:---:|:---:|:---:|
| Calm water | 550 | 10.95 | 0.48 | 0.44 | 0.39 | 3.16 | 0.68 |
| Bad weather | 289 | 5.75 | 0.26 | 0.24 | 0.20 | 1.67 | 0.36 |
| TOTAL | 839 | 16.7 | 0.74 | 0.68 | 0.59 | 4.83 | 1.04 |

*3.2. Concentrations Contributed by Ship Emissions Based on the Air Quality Model*

For generation of the wind field, Calmet has a micrometeorology module that describes the characteristics of the boundary layer on land and on water. Therefore, the following data were required to run the Calmet meteorological model (the meteorological model is described in Supplementary Material).

In order to evaluate exceedances and/or non-compliances of all the pollutants from ships, the annual mean concentrations for each inland grid cell were compared to reference standards and guidelines (WHO and EU).

The highest concentrations when shipping emissions were included (considering all grid cells of the domain) were as follows: for $SO_x$: 29.8, 12.9, and 1.3 µg/m$^3$ (percentile 99.73 hourly, percentile 99.18 daily, and annual average, respectively) for the La Línea de la Concepción area; for $NO_x$: 8.7 to 99.6 µg/m$^3$ (annual average and percentile 99.79 hourly, respectively) in the western area of the Strait; for $PM_{10}$: 0.43 µg/m$^3$ annual average and 0.86 hourly percentile 90.41 and 0.21 µg/m$^3$ annual average for $PM_{2.5}$ in the western area of the Strait.

The results showed (µg/m$^3$) no exceedances of the EU annual limit standard for $SO_x$, $NO_x$, $PM_{2.5}$, or $PM_{10}$. Figure 5 shows how the emissions from maritime transport contributed to the average hourly, daily and annual emission levels of $SO_x$, $NO_X$, $PM_{2.5}$ and $PM_{10}$ in the Algeciras Bay.

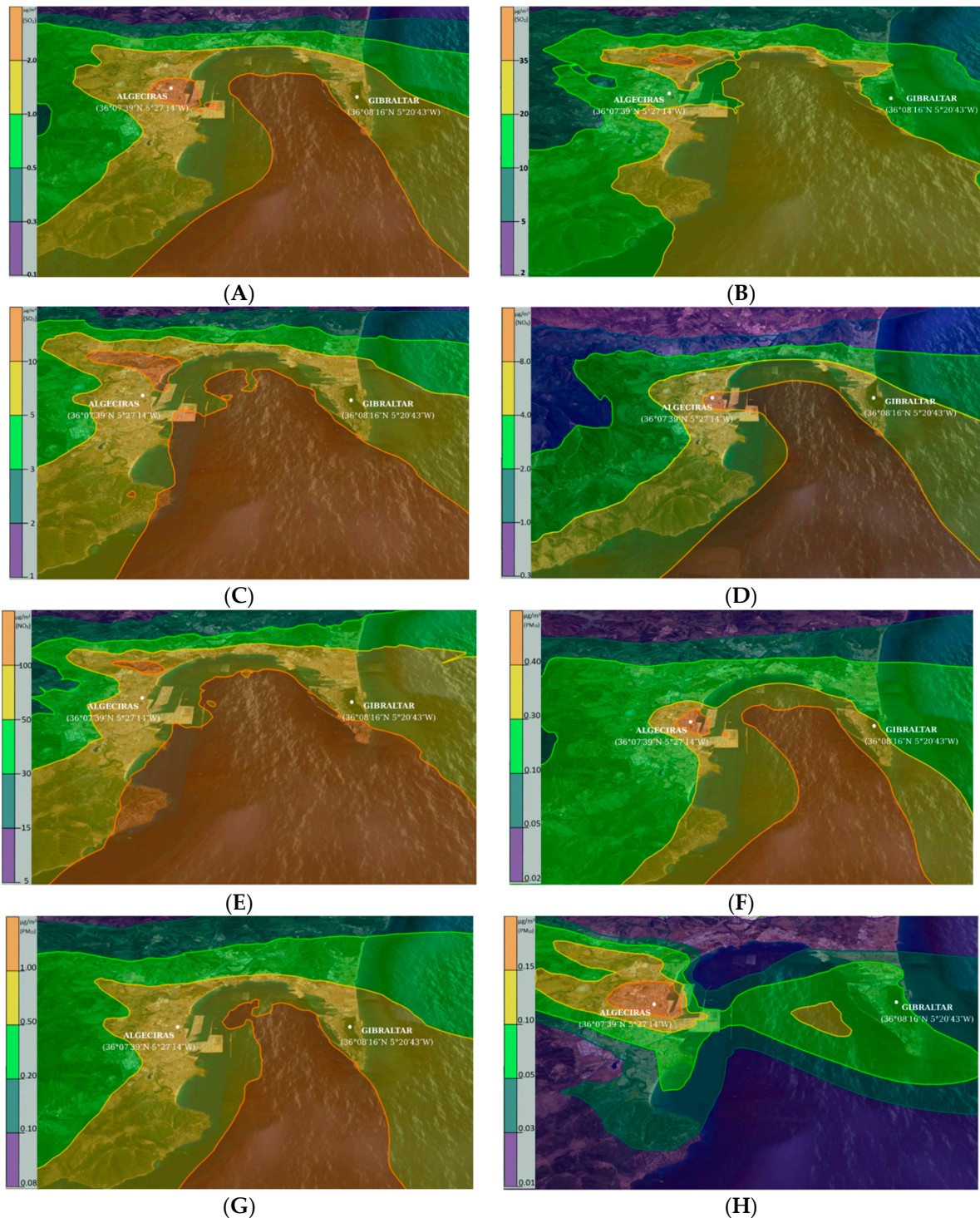

**Figure 5.** Contribution of emissions from maritime transport to mean annual emission levels ($\mu g/m^3$) in Algeciras Bay. (**A**) Mean annual $SO_2$. (**B**) Mean hourly $SO_2$ values, percentile 99.73. (**C**) Mean daily $SO_2$ values, percentile 99.18. (**D**) Mean annual $NO_x$. (**E**) Hourly $NO_x$ values, percentile 99.79. (**F**) Mean annual $PM_{10}$. (**G**) Mean daily $PM_{10}$ values, percentile 90.41 (**H**) Mean annual $PM_{2.5}$.

## 3.3. Comparison with Actual Values Measured at Air Quality Stations

To calculate the contribution of maritime traffic emissions to air quality levels in the study area, the results obtained with the CALPUFF dispersion model were compared to the RVCCAA air quality stations. This analysis has been performed only for the pollutants measured at the stations: $SO_x$, $NO_2$, $PM_{10}$, $PM_{2.5}$, and CO.

The annual average levels recorded at the stations represent the sum of the contributions of all the emission sources in the environment (industry, road traffic, maritime traffic, domestic sector, etc.). Therefore, the difference between the value measured at the station and the value simulated by the dispersion model for the contribution of maritime traffic corresponds to the contribution of other environmental sources.

The percentiles are a statistical parameter that represents hourly or daily values. This means that the measure of the percentile recorded at the station is not the sum of the contributions of the percentiles of all sources of the environment, as is the case in the annual average. Tables 3–5 present the actual results measured at air quality stations compared to the results of the model developed for the simulation of the dispersion of emissions from maritime traffic.

**Table 3.** Compared $SO_x$ results of both simulated and measured marine traffic ($\mu g/m^3$), 2017.

| Air Quality Monitoring Stations | Annual Average | | Percentile 99.73 1 h | | Percentile 99.18 24 h | |
|---|---|---|---|---|---|---|
| | (1) | (2) | (1) | (2) | (1) | (2) |
| Algeciras EPS | 12 | 3 | 52 | 15 | 26 | 7 |
| La Línea | 11 | 1 | 49 | 21 | 23 | 7 |
| Los Barrios | 8 | 1 | 37 | 28 | 22 | 8 |
| E1: Colegio Los Barrios | 12 | 1 | 46 | 18 | 26 | 6 |
| E2: Alcornocales | 8 | 0 | 34 | 2 | 22 | 1 |
| E3: Colegio Carteya | 7 | 0 | 33 | 10 | 21 | 3 |
| E4: Rinconcillo | 11 | 1 | 48 | 30 | 20 | 13 |
| E5: Palmones | 12 | 1 | 39 | 24 | 22 | 9 |
| E6: Estación de FFCC San Roque | 7 | 0 | 74 | 10 | 29 | 3 |
| E7: El Zabal | 12 | 1 | 42 | 17 | 24 | 5 |
| Cortijillos | 9 | 1 | 93 | 17 | 48 | 5 |
| Campamento | 8 | 1 | 89 | 21 | 40 | 6 |
| Economato | 10 | 1 | 87 | 21 | 42 | 5 |
| Escuela de hostelería | 6 | 0 | 45 | 13 | 23 | 3 |
| Guadarranque | 21 | 1 | 217 | 20 | 95 | 7 |
| Madrevieja | 4 | 0 | 75 | 10 | 27 | 3 |
| Puente Mayorga | 13 | 1 | 111 | 25 | 52 | 6 |
| San Fernando | 9 | 0 | 15 | 1 | 14 | 0 |
| Río San Pedro | - | 1 | - | 1 | - | 0 |
| Av. Marconi | 4 | 9 | 8 | 1 | 7 | 0 |
| Prado del Rey | 5 | 0 | 18 | 0 | 15 | 0 |
| Arcos | 3 | 0 | 8 | 1 | 22 | 0 |
| Cartuja | 4 | 0 | 10 | 1 | 7 | 0 |
| Jerez Chapín | 7 | 0 | 13 | 1 | 11 | 0 |
| Limit value National Decree 102/2011 | 20 | | 350 | | 125 | |

Note: (1) Measured. (2) Marine traffic simulated.

**Table 4.** Compared $NO_x$ and $NO_2$ results of both simulated and measured marine traffic ($\mu g/m^3$), 2017.

| Air Quality Monitoring Stations | Annual Average $NO_x$ | | Annual Average $NO_2$ | | Percentile 99.79 1 h Annual Average $NO_2$ | |
|---|---|---|---|---|---|---|
| | (1) | (2) | (1) | (2) | (1) | (2) |
| Algeciras EPS | 44 | 9 | 33 | 9 | 119 | 60 |
| La Línea | 35 | 5 | 23 | 4 | 105 | 76 |
| Los Barrios | 22 | 3 | 16 | 2 | 89 | 79 |
| E1: Colegio Los Barrios | 18 | 1 | 11 | 1 | 71 | 36 |
| E2: Alcornocales | 10 | 0 | 8 | 0 | 53 | 8 |
| E3: Colegio Carteya | 20 | 1 | 14 | 1 | 99 | 30 |
| E4: Rinconcillo | 39 | 5 | 23 | 5 | 110 | 100 |
| E5: Palmones | 38 | 4 | 24 | 4 | 115 | 71 |
| E6: Estación de FFCC San Roque | 26 | 1 | 16 | 1 | 98 | 26 |
| E7: El Zabal | 33 | 3 | 22 | 3 | 108 | 67 |
| Cortijillos | 25 | 2 | 16 | 2 | 113 | 39 |
| Campamento | 19 | 4 | 12 | 3 | 88 | 68 |
| Economato | 17 | 3 | 10 | 2 | 58 | 58 |
| Escuela de hostelería | 28 | 1 | 15 | 1 | 84 | 35 |
| Guadarranque | 31 | 3 | 22 | 3 | 126 | 66 |
| Madrevieja | 17 | 1 | 12 | 1 | 87 | 31 |
| Puente Mayorga | - | 4 | - | 3 | - | 71 |
| San Fernando | 18 | 0 | 13 | 0 | 77 | 7 |
| Río San Pedro | 18 | 0 | 13 | 0 | 76 | 6 |
| Av. Marconi | 25 | 0 | 15 | 0 | 85 | 7 |
| Prado del Rey | 11 | 0 | 6 | 0 | 17 | 1 |
| Arcos | 11 | 0 | 7 | 0 | 28 | 2 |
| Cartuja | 17 | 0 | 10 | 0 | 63 | 4 |
| Jerez Chapín | 28 | 0 | 18 | 0 | 110 | 4 |
| Limit value National Decree 102/2011 | 30 | | 40 | | 200 | |

Note: (1) Measured. (2) Marine traffic simulated.

The results obtained after the application of the dispersion model are shown in Table 3, showing the annual mean, the 99.18 percentile daily, and the 99.73 percentile of the mean hourly values of the emission of $SO_x$.

Regarding the 99.73 percentile of the average hourly levels of $SO_x$ emission caused by maritime traffic emissions, it should be noted that the maximum value achieved at stations of the Air Quality Network was 29.8 $\mu g/m^3$ at station E4: Rinconcillo. Moreover, in the receptors located in the inhabited areas of the study area, the highest value reached was 23.2 $\mu g/m^3$ in the receptor located in the town of La Línea de la Concepción. In both cases, these receptors are located close to the coastline and therefore close to the emission sources. Note that both values are far from the limit value of 350 $\mu g/m^3$ established in Royal Decree 102/2011.

The measure of the percentile recorded at the station is not the sum of the contributions of the percentiles of all sources of the environment, which means that the maximum impact caused by the source will depend on the direction of the wind. This justifies the differences

of measures between the El Riconcillo and Economato stations under southerly wind conditions (Table 4).

**Table 5.** Compared $PM_{10}$ and $PM_{2.5}$ results of simulated and measured marine traffic ($\mu g/m^3$), 2017.

| Air Quality Monitoring Stations. | Annual Average $PM_{10}$ | | Percentile 99.41 24 h $PM_{10}$ | | Annual Average $PM_{2.5}$ | |
|---|---|---|---|---|---|---|
| | (1) | (2) | (1) | (2) | (1) | (2) |
| Algeciras EPS | 27 | 0.4 | 42 | 0.9 | 9 | 0.2 |
| La Línea | 30 | 0.3 | 43 | 0.7 | 24 | 0.1 |
| Los Barrios | 20 | 0.2 | 27 | 0.5 | 16 | 0.1 |
| E1: Colegio Los Barrios | 21 | 0.1 | 31 | 0.3 | - | 0.0 |
| E2: Alcornocales | 18 | 0 | 27 | 0.1 | 8 | 0.0 |
| E3: Colegio Carteya | 23 | 0.1 | 36 | 0.3 | - | 0.0 |
| E4: Rinconcillo | 25 | 0.3 | 39 | 0.8 | - | 0.1 |
| E5: Palmones | 26 | 0.2 | 40 | 0.6 | - | 0.0 |
| E6: Estación de FFCC San Roque | - | 0.1 | - | 0.2 | - | 0.0 |
| E7: El Zabal | 28 | 0.2 | 43 | 0.5 | - | 0.0 |
| Cortijillos | - | 0.1 | - | 0.4 | - | 0.0 |
| Campamento | - | 0.2 | - | 0.6 | - | 0.0 |
| Economato | - | 0.1 | - | 0.5 | - | 0.0 |
| Escuela de hostelería | - | 0.1 | - | 0.3 | - | 0.0 |
| Guadarranque | - | 0.2 | - | 0.6 | - | 0.0 |
| Madrevieja | - | 0.1 | - | 0.3 | - | 0.0 |
| Puente Mayorga | - | 0.2 | - | 0.6 | - | 0.0 |
| San Fernando | 23 | 0 | 34 | 0 | 10 | 0.0 |
| Río San Pedro | 30 | 0 | 44 | 0 | - | 0.0 |
| Av. Marconi | 25 | 0 | 38 | 0 | 8 | 0.0 |
| Prado del Rey | 28 | 0 | 41 | 0 | - | 0.0 |
| Arcos | 28 | 0 | 45 | 0 | - | 0.0 |
| Cartuja | 30 | 0 | 48 | 0 | - | 0.0 |
| Jerez Chapín | 27 | 0 | 44 | 0 | - | 0.0 |
| Limit value National Decree 102/2011 | 40 | | 50 | | 25 | |

Note: (1) Measured. (2) Marine traffic simulated.

As with $SO_x$, the stations most affected by $NO_x$ and $NO_2$ emissions from maritime traffic are located in the Bay of Algeciras. In terms of annual averages, the stations that reflect a greater contribution of maritime traffic (in $\mu g/m^3$) are Algeciras EPS, E4: Rinconcillo, and La Línea, with Algeciras EPS recording the greatest contribution—9 $\mu g/m^3$ of $NO_x$ and $NO_2$—against the actual values measured at that station of 44 and 33 $\mu g/m^3$, respectively. For the percentile 99.79 hourly $NO_2$, note that this is the maximum hourly number of 18 (since the Royal Decree 102/2011, allows 18 exceedances). The stations most affected in this case are E4: Rinconcillo, Los Barrios, and La Línea. E4: Rinconcillo is the station with the greatest impact, with 100 $\mu g/m^3$ (compared to 110 $\mu g/m^3$ measured by the station). As Table S7 (Supplementary Material) shows, there are some stations for which the levels of emissions caused by maritime traffic (simulated with CALPUFF) are very similar to the actual measurements, so it can be concluded that there may be time episodes in which maritime traffic is exclusively responsible for the high values (for example, in E4:

Rinconcillo and Economato). This can happen with southerly winds, during which neither station would be affected by emissions from other sources of the environment.

Uncertainties in particle measurement from combustion are higher than for other pollutants. The $PM_{10}$ and $PM_{2.5}$ emission factor values applied in this study depend on the sulphur content of the fuel and the state of combustion. In this study, the optimum combustion conditions and the most favourable conditions for sulphur content in the fuel were assumed. This could be the reason why the particulate emission factor values provided by maritime transport are so low.

In the east and west domains, primary shipping emissions contributed 1 $\mu g/m^3$ per year to the $PM^{2.5}$ in the atmosphere and 9.2 $\mu g/m^3$ to the $SO_x$ concentrations. Figure 6 shows the evolution of the average hourly values of $SO_x$ emissions.

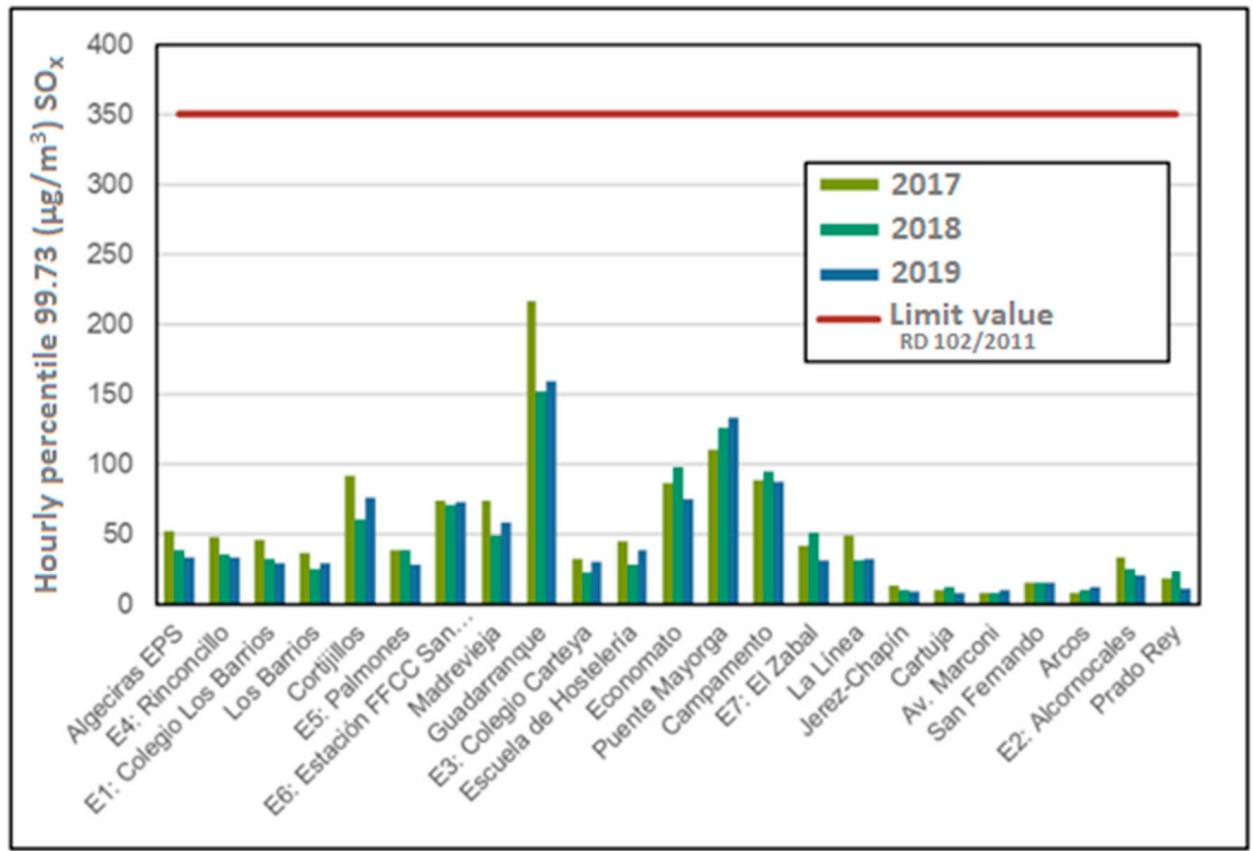

**Figure 6.** Evolution of 99.73 percentile of the average hourly values of $SO_x$ emissions.

The contributions in the area of the Strait of Gibraltar, were more than 90% for $NO_x$ and $SO_x$, 25–50% for $PM_{2.5}$, and 20–35% for $PM_{10}$.

### 3.4. Discussion

A comparison of the results obtained for $SO_x$, $NO_x$, $PM_{10}$, $PM_{2.5}$, and CO pollutants with the CALPUFF dispersion model at RVCCAA air quality stations is then made against the actual values measured at those stations to calculate the contribution of maritime traffic emissions to air quality levels in the study area.

Similar results have been obtained for other similar studies carried out in the Mediterranean Sea [30–32].

As regards the annual average levels recorded at the stations, it must be noted that they represent the sum of the contributions of all emission sources in the environment (industry, road traffic, maritime traffic, domestic sector, etc.).

Air quality impact assessment methodology leads to inevitable uncertainties, and all steps from the emissions model to air pollution were included in this study. Nonetheless, some overestimations may occur, thus results must be analysed with caution. Regarding the exposure assessment, the most significant uncertainties came from the choice of the pollutants, the general shape of the Emissions factors values and their applicability from each place and the population exposed. The air pollution is a known complex mix of gases, but in this case, only those from shipping were studied. The nature of emissions from ships describes $PM_{2.5}$ fraction as the most widely used and accepted indicator, but in this study, the air quality impacts associated with exposure to $NO_2$ were also evaluated.

Therefore, the difference between the measured value at the station and the simulated value with the dispersion model for the contribution of maritime traffic would correspond to the contribution of the other sources.

A series of conclusions can be drawn from their analysis. The levels of $SO_x$ emissions caused by maritime traffic emissions evaluated at stations of the Air Quality Monitoring and Control Network, inhabited areas, and areas of ecological interest remain below the limits established in Royal Decree 102/2011 for the protection of human health and ecosystems. The contribution of emissions from ships to the average annual levels of emission of $SO_x$ is very small compared to the limit value of 20 $\mu g/m^3$ established by this decree. As can be seen, the maximum value recorded in discrete receptors located in ecological areas is 1.3 $\mu g/m^3$ in the "Eastern Strait SCI" receptor.

Finally, regarding the influence on the daily averages of $SO_x$, the maximum percentile caused by the ships is 30 $\mu g/m^3$ at station E4: Rinconcillo for the stations of the Air Quality Network and 7.3 $\mu g/m^3$ in the receptor located in the municipality of La Línea de la Concepción for receptors located in inhabited areas. These values are far from the limit value of 125 $\mu g/m^3$ established in the legislation in force.

With regards to particulate matter $PM_{10}$ and $PM_{2.5}$, it should be noted that the levels found at air quality stations as a result of emissions from maritime traffic are lower than expected compared to the limit values established in Royal Decree 102/2011 for both the annual average (40 $\mu g/m^3$) and the 90.41 daily percentile (50 $\mu g/m^3$) of $PM_{10}$ and the annual average of $PM_{2.5}$ (25 $\mu g/m^3$). The stations that present the greatest contribution from the maritime sector (in $\mu g/m^3$) to the average annual levels of emission of $PM_{10}$ are Algeciras EPS, E4: Rinconcillo, and La Línea. However, in all cases, the simulated values are below 0.5 $\mu g/m^3$ compared to the actual levels measured at stations of around 18–30 $\mu g/m^3$. For the 90.41 daily percentile of $PM_{10}$, this is the maximum daily number of 35 (since the Royal Decree 102/2011 allows 35 exceedances of the daily limit value of 50 $\mu g/m^3$). The stations most affected in this case are also Algeciras EPS, E4: Rinconcillo, and La Linea, presenting simulated values in each case below 1 $\mu g/m^3$ (the actual values measured at these stations are between 27–48 $\mu g/m^3$). Finally, after analysing the results of the average annual levels of $PM_{2.5}$ in the air quality stations, those most affected by maritime traffic were found to be the same as for $PM_{10}$ (including also Los Barrios), with estimated levels at all these stations below 0.3 $\mu g/m^3$ compared to the actual levels measured between 8 and 24 $\mu g/m^3$. As an example, Figure 5 shows the $SO_x$ concentrations. Palmones was identified as a major hotspot with mean daily ship-sourced concentrations >215 $\mu g/m^3$. On the other hand, the highest concentrations of $PM_{10}$ from ships reached 8.5 $\mu g/m^3$ inside the Strait.

### 3.5. Uncertainties and Limitations

The complexity of chemical transport models is very significant, and for this reason, it is difficult to specify all the sources of uncertainties; this is due to the appropriateness of the meteorological data, emission inventory, and the imperfections of chemical mechanisms and physical processes in the modelling system corresponding to the emissions inventory, which produces a major cause of uncertainty. In the case of the SENEM model, unlike other known models, this uncertainty is limited to emission factors (specific fuel oil consumption, fuel type, fuel sulphur content, etc.). Power prediction (weather contributions, fouling, squat,

sea currents, auxiliary engine power profiles, engine load estimation, power transmission, propeller properties), is controlled in the SENEM model.

On the other hand, uncertainties concerning emission factors may be larger for products of incomplete combustion, such as CO, non-methane volatile organic compounds (NMVOCs), OC, EC, and $CO_2$ or $NO_x$ because these are strongly related to engine load, engine generation, and service history. All these circumstances have been taken into account in this study [13].

Keeping the uncertainties of the atmospheric dispersion simulations in mind, efforts were made to run the CALPUFF model as accurately and in as detailed a manner as possible. Moreover, although it has been possible to identify variations in emissions and concentrations near port areas, the resolution that was used was too coarse to make a detailed analysis of emissions and concentrations inside port areas. The CALPUFF model considers the $O_3$ loss by $NO_x$ titration, the sunlight effects, and the $NO_x$-to-VOC ratio that promotes $O_3$ production, which is an approximation allowing for the minimization of the effects of the non-linear $O_3$ chemistry.

## 4. Conclusions

Following the implementation of the model and the subsequent analysis of the results obtained, the following conclusions are drawn:

- The levels of pollutants recorded at existing air quality stations in the province of Cádiz during the period 2017–2019 are below the limit values set in the Spanish Royal Decree 102/2011; European Commission, 2018 [33]; and WHO, 2018 [34] concerning the improvement of air quality for the protection of human health, vegetation, and ecosystems.
- The "Bahía de Algeciras Industrial Zone (ES0104)" recorded the highest levels and the greatest differences from the other zones ("Zona Bahía de Cádiz (ES0124)"; "Zonas rurales (ES0123)") for $SO_x$.
- The pollutants presenting the highest levels of emission as a result of maritime traffic in the area of the Bay of Algeciras and the Strait of Gibraltar are $SO_x$ and $NO_2$.
- In the specific case of the annual average values of $SO_x$ and $NO_2$, the contribution of maritime traffic to air quality levels may be around 20–25% for the most susceptible receptors, especially in the area of Algeciras. In the case of the percentile analysis, maritime traffic can have a significantly greater impact, especially in the case of $NO_2$.

**Supplementary Materials:** The following supporting information can be downloaded at: https://www.mdpi.com/article/10.3390/su141912507/s1, Figure S1: RVCCAA stations present in the province of Cádiz; Figure S2: Ozone station locations; Figure S3: Routes and line points studied; Figure S4: Location of the weather stations considered in the dispersion model; Table S1: Characteristics of the air quality monitoring stations "Algeciras Bay industrial area"; Table S2: Characteristics of the air quality monitoring stations "Cádiz Bay area"; Table S3: Characteristics of the air quality monitoring stations "rural area"; Table S4: Baseline Emission Factors (g/kWh) for main and auxiliary engines; Table S5: Air quality stations that record ozone values; Table S6: Main and auxiliary engines power and load factors by ship type in cruising mode; Table S7: Specification WRF model. References [35–47] are cited in the supplementary materials.

**Author Contributions:** J.M.-G. and V.D.-G.: conceptualization, methodology, writing—original draft, funding acquisition, supervision. R.R.-M. and F.C.-C.: methodology, formal analysis. Y.A.-S. and E.P.-V.: investigation. R.A.O.N.: air quality model. M.C.M.A.-F. and S.I.V.S.: conceptualization and data curation. V.D.-G.: head of the project. All authors have read and agreed to the published version of the manuscript.

**Funding:** This research was funded by Consejería de Salud (Andalusian Government) and FEDER, grant number PI-0094-2017. Project "The influence of Maritime Traffic on Human Health. Proposal of a new model for calculating mortality and morbidity in the province of Cádiz (Spain)". And funded by National funds through FCT/MCTES (PIDDAC), grand number LA/P/0045/2020 (ALiCE) and UIDB/00511/2020-UIDP/00511/2020 (LEPABE). And also funded by FEDER funds through

COMPETE2020-Programa Operacional Competitividade e Internacionalização (POCI) and by national funds (PIDDAC) through FCT/MCTES, grand number POCI–01–0145–FEDER–032201 (EMISSIP).

**Institutional Review Board Statement:** Not applicable.

**Informed Consent Statement:** Not applicable.

**Data Availability Statement:** Not applicable.

**Conflicts of Interest:** The authors declare that they have no known competing financial interests or personal relationships that could have appeared to influence the work reported in this paper.

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
