# Peer review of "The Influence of Emissions from Maritime Transport on Air Quality in the Strait of Gibraltar (Spain)"

_sustainability, doi:10.3390/su141912507_

Round 1
Reviewer 1 Report (New Reviewer)
General comments
Ship traffic over the ocean is responsible for gaseous and particulate emissions that have a significant effect on the air quality. In addition, these emissions play a role in the physicochemical processes in the atmosphere involved with the global climate. That is why we need data of gas and particulate emissions to be able to establish accurate climate future scenario and hence, the results proposed in the present manuscript is then of great interest. However, if the experimental results could be useful to our community, they suffer from a lack of description and the comparison with model simulations brings a lot of questions. That is why before publication, I recommend a major revision taking into account my comments reported below.
Major concerns
The main results of the present manuscript are based on a comparison between aerosol concentration data and numerical model simulations. The large uncertainties on shipping emission inventories and the models used for numerical simulations would need more discussion and should prevent a too rapid conclusion (i.e., "The incidence of maritime traffic emissions on the levels of particle emission... is insignificant..."). It should be the aim of the present manuscript to assess the reliability of their shipping emission inventories and the accuracy of the comparison with the air quality model ouptuts. In contrast with the three lines reported in Section 3.5 dedicated to the "Uncertainties and limitations", which are too short and vague, the authors should then produce a strong effort to detail the possible influence of some of them on the results and hence, to explain the substantial disagreement shown in their manuscript between the experiments and model calculations, as reported in Table 12.
- Two much numerical models are used for the study (SENEM, CALPUF, RVCCAA ...) and it is a bit confusing. We could have a redundancy between all these models...
- The model used for the aerosol concentrations reported in Table 12 seems to provide very low values. This makes think to a probable inconsistency of emission inventories or modeling resolution. It seems like the model only uses source terms provided by default using the high resolution European inventory (for instance ECMWF). The authors should try to find out what is going wrong in their simulations? Inventories compiled from ship activity records and activity-based emission factors for different ship types could be more explained.
- Estimation of the contribution of maritime traffic using the difference between the value measured at the station and the value simulated with the dispersion model should be discussed. First of all, uncertainties on the model dispersion could be more detailed since they affect the difference between the measured value at the station and the simulated value with the dispersion model for the contribution of maritime traffic, which corresponds to the contribution of the other sources. The authors could reduce uncertainties by comparing their results on the contribution of the other sources to those found in the literature on aerosol concentrations of pollutants in other Mediterranean locations (e.g., Bardouki et al., 2003; 2004; Eleftheriadis et al., 2006; Piazzola et al., 2016).
- It could be more convincing to study specifically two or three particular meteorological episodes for which the authors could provide a temporal survey of the main meteorological data, as wind speed and direction, relative humidly and maybe also the height of the MABL. and relate them to the measured aerosol concentrations at each receptor.
-The authors conclude that the incidence of maritime traffic emissions on atmospheric particle concentrations is insignificant. Even though the percentage of ship emissions on total atmospheric pollutants is less than 20 %, does it mean that it is really insignificant. And insignificant for what? Health, air quality, global warming?
Minor comments
-pp. 1 "emission " not " immission".
- Fig. 6 and Fig. 7 are missing.
- result of emissions from maritime traffic are insignificant compared with the limit values established in Royal Decree 102/2011..Too strong.
References
Bardouki, H., Liakakou, H., Economou, C., Sciare, J., Smolik, J., Zdimal, V., Eleftheriadis, K., Lazaridis, M., Dye, C., Mihalopoulos, N.(2003). Chemical composition of size resolved atmospheric aerosols in the eastern Mediterranean during summer and winter. Atmos. Environment, vol. 37, n°2, 195-208, doi:10.1016/S1352-2310(02)00859-2.
Eleftheriadis, K.,_, I. Colbeck, I., C. Housiadaa, C., M. Lazaridis, M., N. Mihalopoulos, N., C. Mitsakou, C., J. Smolık, J., and V. Zdımal, V. (2006). Size distribution, composition and origin of the submicron aerosol in the marine boundary layer during the eastern Mediterranean ‘‘SUB-AERO’’ experiment. Atmospheric Environment, 40 6245–6260.
Piazzola, J. Mihalopoulos, N. Canepa, E. Tedeschi, G. Prati, P. Zarmpas, P. Bastianini, M. Missamou, T. et L. Cavaleri (2016), Characterization of aerosols above the Northern Adriatic Sea: case studies of offshore and onshore wind conditions, Atmospheric Environment n° 132. pp. 153–162, doi:10.1016/j.atmosenv.2016.02.044.

Author Response
Please see the attachment.

Reviewer 2 Report (New Reviewer)
Comments to the Author
Review of Article “The Influence of Emissions from Maritime Transport on Air Quality in the Strait of Gibraltar (Spain)” by Vanessa Durán-Grados, Rubén Rodríguez-Moreno, Fátima Calderay-Cayetano, Yolanda Amado-Sánchez, Emilio Pájaro-Velázquez, Rafael A.O.Nunes, Maria C.M.Alvim-Ferraz, Sofia I.V.Sousa, Juan Moreno-Gutiérrez.
In my opinion, it is an interesting study. It is a good attempt to investigate the impact of shipping and marine activity in an interesting region in terms of marine traffic – the Gibraltar straits. For the analysis they used a variety of data (stations and models). Of course the uncertainty and limitations of models are important but this analysis can provide (a) significant qualitative results regarding the air quality and (b) the general context to further understand the impact of marine activity in the strait of Gibraltar.
I believe that it would be valuable if the authors provide an extension of literature in introduction section regarding the impact of pollution on human health and environment. Also, I suggest the authors reorder a bit the section of “Introduction and Background”. In my opinion, it would be valuable a short discussion regarding the impact of climate changes and marine activity and the impact of African dust Episodes in the air quality of the region of Gibraltar. As a general comment, I suggest the authors follow a separate numbering over the main manuscript and supplementary material. It is confusing for the reader to follow the text when the numbering of figures and tables are mentioned in two different texts. I recommend the authors keep the numbering in the form of “Figure 1, Figure 2 e.t.c” for the main manuscript and “Supplementary Figure 1 or SFigure 1, Supplementary Figure 2 or SFigure 2 e.t.c” for the supplementary material. In some places I suggest the authors discuss a bit more and provide some more literature. Finally, I recommend the authors take under consideration the following comments:
Abstract
1. I suggest the authors include in the abstract the period that this study focuses on (the year 2017).
Background and Introduction
1. In lines 27 – 31: I suggest the authors include some references regarding these points.
2. In my opinion, the authors should provide some literature in lines 25-26 and in lines 27-30 regarding the impact of air pollution on human health and environmental quality. Also I suggest the authors provide some references regarding the Epidemiological studies that they discuss in lines 34-35.
3. I recommend a short redirection of the “Background and Introduction”. In my opinion, the authors should provide at first the literature regarding the background of this work and secondly present the elements of their study, the objectives and the main contribution to scientific community. I suggest to transfer the lines 39-46 in the last section of the “Background and Introduction” section.
4. I suggest the authors also provide some literature elements regarding the air quality and shipping in the context of climate change. In my opinion, two or three sentences could provide some elements regarding this region which is located in a susceptible domain for the climate change.
5. As a general comment I suggest a short discussion regarding the impact of African dust episodes. I believe that it would be valuable a short discussion (adding literature) regarding the impact of dust episodes in combination with marine activities and their influence on the air quality and human health in straits region.
6. In my opinion, it would be significant the authors provide some elements regarding the limitations of this study in this section too.
Methodology
1. In my opinion, you should provide one or two sentences where you present the variables (the name of pollutants) that you study in this work.
2. Could you please correct the numbering of the Figures all over the manuscript?
The first figure into the text is “Figure 3. - Strait of Gibraltar – Area studied.”. Could you please replace this with “Figure 1. - Strait of Gibraltar – Area studied.” considering it as the first figure of this manuscript?
Lines 131 -133 the authors said that “Therefore, for the definition of the scenario to be simulated, emissions are considered under normal conditions for the months of March to October and emissions under adverse weather conditions for the months of January, February, November and December.” If I understand correct you chose this scenario because it is of great importance regarding the impact of wind speed on shipping affecting the air quality of the Straits. Could you please provide some elements regarding this point from a meteorological aspect? For example, is it an atmospheric circulation pattern that provide this wind pattern? Is the frequency of east and west winds significantly lower during the period from November to February compared to the other period of the year (from March to October)? Possibly the combination of local atmospheric circulations or synoptic conditions in winter period in combination with the topography of the straits affect the winds over the straits during the winter. You have already provided some information regarding this point in the supplementary material (in the CALMET meteorology).
Could you please discuss a bit these points?
3. Please see my previous comment (comment 3) and correct the numbering of Figure 4.
4. In Figure 4 (line 157) the wind rose shows that the dominant wind directions are the west and east during November – February period. The authors said that this wind pattern leads to “the loss of one knot to maintain the same speed”. The continental region that the main population lives is in the south and north of the straits. The west and east winds, of course, leads to the “power delivered increasing, as did fuel consumption and emissions”. Is it correct that this wind pattern affects the reduction of pollutants over the area (via the dispersion of pollutants) where the main population lives? In other words, these wind directions affect the shipping but tend to improve the air quality in the north and south of the straits where the main population lives. Could you please discuss this point?
5. Please see my previous comment and correct the numbering of Figure 5 (Comment 5&3).
6. In lines 212-214 the authors said that “The total results for the Strait of Gibraltar can be found in Table 8. It shows the emissions of PM2.5, SOx, and NOx for eight months of calm water conditions and four months 213 in which the weather was bad.” The eight months with calm water is the period from March to October and the months with bad weather is the period from November to February? Could you please clarify?
7. Please see my previous comment regarding the numbering of Figures.
If I understand correctly Figure 7 (line 233) is the Figure 7 from the supplementary material. I suggest the authors move this figure in the main text (manuscript) and change the numbering of the Figures all over the text.
8. In lines 287-288 the authors said that “This can happen with southerly winds, where neither station would be affected by emissions from other sources of the environment”. It is possible the southerly wind circulation to affect the air quality due to the transfer of the African dust. Could you please discuss this point? Are there some dust episodes that affect the stations? Of course, in this analysis, the authors separate the impact of marine activity on the air quality but a short discussion regarding the impact of dust episodes is valuable because it is also an important factor that affects the air quality over the region.
Minor comments
1. In line 45 please add a gap between the word “CAPLUFF” and the parenthesis. Please replace with “CALPUFF (both are described…)”.
2. In line 105 please remove the second full stop.
3. In line 163 please add a gap in “case(Figure1)” and replace with “case (Figure1)”.
4. In line 167 you have one extra gap.
5. In line 343 please remove the “ar”
6. In lines 344 please replace the full stop.
7. In line 360 please add a full stop and replace with the sentence “…..especially in the area of Algeciras.”
8. Please replace the numbering of the References with the correct font.
Round 2
Reviewer 1 Report (New Reviewer)
The article is now rok for publication
Reviewer 2 Report (New Reviewer)
In my opinion, the revised version of the article with title “The Influence of Emissions from Maritime Transport on Air Quality in the Strait of Gibraltar (Spain)” by Vanessa Durán-Grados, Rubén Rodríguez-Moreno, Fátima Calderay-Cayetano, Yolanda Amado-Sánchez, Emilio Pájaro-Velázquez, Rafael A.O.Nunes, Maria C.M.Alvim-Ferraz, Sofia I.V.Sousa, Juan Moreno-Gutiérrez is improved compared to the previous one.
The authors have answered the reviewer's comments and also have followed the majority of suggestions.
I suggest the publication of the article in the "Sustainability " Journal.
This manuscript is a resubmission of an earlier submission. The following is a list of the peer review reports and author responses from that submission.
Round 1
Reviewer 1 Report
This research used the Ship´s Energy and Emissions Model (SENEM) and the California Puff air quality model (CALPUFF) to estimate the impact of NOx, SOx and particulate matter (PM2.5) on air quality in the Strait of Gibraltar (Spain). Their results showed that the Algeciras Bay Industrial Zone had the highest levels of pollutants and the Palmones area was identified as the main hotspot. The following issues could be addressed by the authors before publication to increase the quality of the manuscript.
Emissions are related to the required energy, the emission factor and the fuel correction factor. Therefore, the authors could further discuss how the fuel correction factor was considered in the emissions calculation. Because the types of fuel used by the ships were marine diesel oil (30%), residual oil (62%) and liquified natural gas (8%).
The authors could describe how the computational models take into account the effects of the ship trajectories and wind speeds on the emissions at different locations.
The authors could provide some evidence when the emission levels are similar to actual measurements (e.g. E4: Rinconcillo and Economato, in Table 10) if they were under southerly wind conditions and the site would not be affected by the emissions from other environmental sources.
There are some typos, the article needs to be checked again, for example:
Line 503: …NO2, PM10, PM2,5 y CO.…
Line 526: …Tables 9, 10 and 110 present the actual...
Line 535:...levels of SO2 immission...
Line 551: …Network and 7.3 μg/m3…
Line 566:...As Table 6 shows....
Line 566:...where the levels of emissioncaused...
Author Response
THE INFLUENCE OF EMISSIONS FROM MARITIME TRANSPORT ON AIR QUALITY IN THE STRAIT OF GIBRALTAR (SPAIN).
Dear Editor and Reviewers
The authors would like to thank you for the time taken to review the manuscript. All the comments received have been very constructive and have served to greatly improve the manuscript.
Below we answer one by one the editors' and reviewers' comments.
In general terms, a new supplementary section(2,579 words) has been created. The text has therefore been reduced to 6,948 words.
Some references have been added and some references removed.
The references have been cited according to the rules of the journal
Manuscript and Supplementary material have been vevised by a native speaker
Reviewer 1. Comments and Suggestions for Authors
COMMENT
This research used the Ship´s Energy and Emissions Model (SENEM) and the California Puff air quality model (CALPUFF) to estimate the impact of NOx, SOx and particulate matter (PM2.5) on air quality in the Strait of Gibraltar (Spain). Their results showed that the Algeciras Bay Industrial Zone had the highest levels of pollutants and the Palmones area was identified as the main hotspot. The following issues could be addressed by the authors before publication to increase the quality of the manuscript.
Emissions are related to the required energy, the emission factor and the fuel correction factor. Therefore, the authors could further discuss how the fuel correction factor was considered in the emissions calculation. Because the types of fuel used by the ships were marine diesel oil (30%), residual oil (62%) and liquified natural gas (8%).
Thank you so much for your constructive comment
ANSWER
Table 4 (Supplementary Material) shows the Baseline Emission Factors (g/kWh) values for Main and Auxiliary Engines that has been used for calculation.
All the references below have been added also
Quantification of emissions from ships associated with ship movements between ports in the European Community. UK: Report prepared for the european Commission.ENTEC.2002. http://ec.europa.eu/environment/air/pdf/chapter2_ship_emissions.pdf .
Inventory of U.S. Greenhouse Gas Emissions and Sinks: 1990–2012. USEPA. 2014.
Kristensen H. O. Energy demand and exhaust gas emissions of marine engines. Project no. 2010–56, Emissions be slutnings støtte system, Work Package 2, report nº. 05, September.2012
Cooper, D.; Gustafsson, T. Methodology for calculating emissions from ships:Update of emission factors, IVl (Swedish environmental research institute).2004
Kunz, P.;Gorse, P. Development of high-speed engines for natural gas operation in tugs. Tugnology ’13.2013
Samulski, M. Estimation of Particulate Matter Emissions Factors for Diesel Engines on Ocean-Going Ships. USEPA .2 0 0 7
Sarvi, A.; Fogelholm, C.J.;Zevenhoven, R. Emissions from large-scale medium-speed diesel engines:Influence of engine operation mode and turbocharger. Fuel Processing technology. 2008, 89, 510–519.
USEPA (2014). Inventory of U.S. Greenhouse Gas Emissions and Sinks: 1990–2012
COMMENT
The authors could describe how the computational models take into account the effects of the ship trajectories and wind speeds on the emissions at different locations.
Thank you so much for your comment. We considere very important and neccesary that this comment appears in the text
ANSWER
Lines 231-234 has been added
The SENEM model uses the Kwon (Known, 2008) method to predict speed loss due to added resistance in abnormal weather conditions (irregular waves and wind), when Equation 1 (Moreno and Duran, 2021) was applied. The Kwon model has the advantage that it is easy and practical to use.
COMMENT
The authors could provide some evidence when the emission levels are similar to actual measurements (e.g. E4: Rinconcillo and Economato, in Table 10) if they were under southerly wind conditions and the site would not be affected by the emissions from other environmental sources.
Thank you so much for your comment. We also considere this comment very important
ANSWER
This paragraph has been added in the text(lines 308-312)
The measure of the percentile recorded at the station is not the sum of the contributions of the percentiles of all sources of the environment, which means that the maximum impact caused by the source will depend on the direction of the wind. This justifies the differences of measures between the El Riconcillo and Economato stations under Southern wind conditions (Table 11).
There are some typos, the article needs to be checked again, for example:
Line 503: …NO2, PM10, PM2,5 y CO.…
It´s now fixed
Line 526: …Tables 9, 10 and 110 present the actual...
It´s now fixed. They have replaced by tables 10, 11 and 12
Line 535:...levels of SO2 immission...
It´s now fixed
Line 551: …Network and 7.3 μg/m3…
It´s now fixed
Line 566:...As Table 6 shows....
It´s now fixed
Line 566:...where the levels of emissioncaused...
It´s now fixed
Reviewer 2 Report
Dear authors,
I just started reading the paper but I stopped in section 2, as I found many writing and editing errors. Serve as examples:
- Line 12 of abstract mentions only three pollutants but next, results from other pollutants are mentioned
- In the introduction, paragraphs of the same topics are mixed: paragraph in line 65 should be joint with paragraph in line 77.
- Lines 70-71 sentence not should be there.
- Some words underlined
- Table 1, 2.. should be well formatted, they are too big and not in accordance with the letter in the manuscript.
Please, revise the full manuscript, format, content and organization, and then resubmit the manuscript.
Author Response
Dear Editor and Reviewers
The authors would like to thank you for the time taken to review the manuscript. All the comments received have been very constructive and have served to greatly improve the manuscript.
Below we answer one by one the editors' and reviewers' comments.
In general terms, a new supplementary section(2,579 words) has been created. The text has therefore been reduced to 6,948 words.
Some references have been added and some references removed.
The references have been cited according to the rules of the journal
Manuscript and Supplementary material have been vevised by a native speaker
Reviewer 2.-
I just started reading the paper but I stopped in section 2, as I found many writing and editing errors. Serve as examples:
COMMENT
- Line 12 of abstract mentions only three pollutants but next, results from other pollutants are mentioned.
Thank you so much for your comment
ANSWER
Only has been replaced by mainly
- COMMENT
In the introduction, paragraphs of the same topics are mixed: paragraph in line 65 should be joint with paragraph in line 77.
ANSWER
It has been corrected
- Lines 70-71 sentence not should be there.
It´s now fixed
- Some words underlined
It´s now fixed
- Table 1, 2.. should be well formatted, they are too big and not in accordance with the letter in the manuscript.
It´s now fixed and placed in the supplementary material section
Please, revise the full manuscript, format, content and organization, and then resubmit the manuscript.
Thank you so much for your comment
ANSWER
All the manuscript has been resctrutured and rewritten
A new Supplementary material section has been added
Reviewer 3 Report
In this manuscript, the authors attempted to analyze the maritime transport emissions on air quality using emission/transport models. The current version of the manuscript contains a lot of redundancies and repetitions, which make it lengthy and difficult to follow. It is not presented in a standard format of a scientific paper, rather it is presented as a technical report without an in-depth discussion of results. It needs rewriting to be published in any journal. Also, the scientific contribution of the study is not clear. For these reasons, I cannot recommend this manuscript for publication in Sustainability.
Some additional comments are as follows:
- The introduction section is written with poor connection and flow. For example, why is it necessary to present Line 50-53 in a separate paragraph? Can’t it be merged with the previous paragraph? Please maintain a logical flow and connection in the introduction section.
- Line 70-71: “Once all of them answered, the article has improved a lot and we hope that it will be 70 to your liking.In the case of the Iberian Peninsula, another study identified…” What does it mean? Did the authors proofread their manuscript before submitting it?
- Line 75-76: A paragraph of a single sentence?
- Line 77-80: Another paragraph of two sentences?
- The introduction section is poorly written without flow and coherence. It needs to be rewritten.
- What are the authors' motivations to conduct this study? The introduction section needs to focus on a critical review of existing studies, identification of gaps, and rationale of the present study. These aspects are lacking in this manuscript. Also, a clear statement of the scientific contribution of this study is needed.
- Line 130-131 Statement of the objective is unnecessary here (methods); It is already stated in Line 119-120 (introduction).
- Tables 1-3 can be moved to supplementary material.
- Line 151-156 and Line 289-293: Same information is presented twice in the methods sections. This manuscript has several such issues, which make it very lengthy and difficult to follow. It needs substantial improvement.
- The manuscript lacks a discussion of the results.
- Present only the key findings and implications of the study in the conclusion section.
Author Response
Dear Editor and Reviewers
The authors would like to thank you for the time taken to review the manuscript. All the comments received have been very constructive and have served to greatly improve the manuscript.
Below we answer one by one the editors' and reviewers' comments.
In general terms, a new supplementary section(2,579 words) has been created. The text has therefore been reduced to 6,948 words.
Some references have been added and some references removed.
The references have been cited according to the rules of the journal
Manuscript and Supplementary material have been vevised by a native speaker
Reviewer 3.-
COMMENT
In this manuscript, the authors attempted to analyze the maritime transport emissions on air quality using emission/transport models.
The current version of the manuscript contains a lot of redundancies and repetitions, which make it lengthy and difficult to follow.
ANSWER
Thank you so much for your comment. The manuscript has been resctrutured and rewritten
COMMENT
It is not presented in a standard format of a scientific paper, rather it is presented as a technical report without an in-depth discussion of results.
We considere very important and constructive comment
ANSWER
A discussion section has been added
COMMENT
It needs rewriting to be published in any journal.
ANSWER
Full agree. Thank you so much. All the manuscript has been resctrutured and rewritten
COMMENT
Also, the scientific contribution of the study is not clear. For these reasons, I cannot recommend this manuscript for publication in Sustainability.
ANSWER
Thank you so much for your comment, the following parragraphs have been added in the introduction section. Lines 38-49.
The reason for this is that the future growth of ship traffic will affect the composition of the atmosphere, contributing to worsening air quality. These emissions are also generated while vessels are at berth and affect not only major ports, but also medium and small-scale ones [6]. International shipping represents around 13% and 12% of total global anthropogenic emissions of NOx and SOx, respectively [7]. Furthermore, PM emissions from international shipping contribute about 3% - 4% of global emissions [8]. Most (87%) of these were attributable to international shipping activity [9], while domestic shipping was responsible for about 9% of total shipping emissions of CO2 and fishing for around 4%. Examining the make-up of the shipping fleet reveals that 55% was generated by container ships, bulk carriers, and oil tankers.
Some additional comments are as follows:
COMMENT
- The introduction section is written with poor connection and flow. For example, why is it necessary to present Line 50-53 in a separate paragraph? Can’t it be merged with the previous paragraph? Please maintain a logical flow and connection in the introduction section.
Thank you so much. We considere very constructive comment
ANSWER
Introduction section has been rewritten
COMMENT
- Line 70-71: “Once all of them answered, the article has improved a lot and we hope that it will be 70 to your liking.In the case of the Iberian Peninsula, another study identified…” What does it mean? Did the authors proofread their manuscript before submitting it?
ANSWER
Sorry . It was an internal correction. It has been deleted.
COMMENTS
- Line 75-76: A paragraph of a single sentence?
- Line 77-80: Another paragraph of two sentences?
- The introduction section is poorly written without flow and coherence. It needs to be rewritten.
Thank you so much for your comment
ANSWERS
Introduction has been rewritten
The introduction section in original manuscript has been reduced by more than 30%
COMMENT
- What are the authors' motivations to conduct this study? The introduction section needs to focus on a critical review of existing studies, identification of gaps, and rationale of the present study. These aspects are lacking in this manuscript.
We considere very important this comment
ANSWER
The following paragraphs were added
Lines 55-66
Taking into account that many factors can influence air quality in port cities and coastal areas, including geography and climate, road, rail and maritime traffic, industrial and residential emissions, but very little is known about the magnitude and effects of air pollution due to marine vessels [11]. The authors of this manuscript were motivated to develop such a study in the Strait of Gibraltar.
On this topic, it is also relevant to take into account the results of a study applied for the Iberian Peninsula[12](Nunes et al.,2020) showing that in the case of the Strait of Gibraltar, CO2, NOx, sulfate, and SOx had the highest values: 1330, 24, 1.03, and 11.6 t yr-1 km-2, respectively. For these reasons, the authors decided to use two different models to those of this study exclusively applied to the Strait of Gibraltar area; SENEM’s own model for calculating emissions and CALPUFF(described in Supplementary material) for air quality.
COMMENT
Also, a clear statement of the scientific contribution of this study is needed.
Thank you so much for your comment
ANSWER
The following parragraph was added (Lines 81-86)
In order to obtain results that are as true to life as possible, this study uses the SENEM model to quantify emissions from ships. This model, unlike other models such as STEEM or STEAM, quantifies the power delivered by propulsion engines taking into account all the parameters which influence the resistance to the ship advancing. The complete procedures are defined in the SENEM model published by the authors of this paper [14].
the following paragraphs were also included(lines 55-66)
Taking into account that many factors can influence air quality in port cities and coastal areas, including geography and climate, road, rail and maritime traffic, industrial and residential emissions, but very little is known about the magnitude and effects of air pollution due to marine vessels [Mueller et al.,2011], The authors of this manuscript generated motivated to develop such a study in the Strait of Gibraltar.
Taking into account also the results of a study(Nunes et al.,2020) showing that in the case of the Strait of Gibraltar, CO2, NOx , sulfate, and SOx had the highest values, 1330, 24, 1.03, and 11.6 t yr-1 km-2, respectively
For these reasons, the authors decided to use two different models to those of this study in the Strait of Gibraltar area. SENEM’s own model for calculating emissions and CALPUFF for air quality
COMMENT
- Line 130-131 Statement of the objective is unnecessary here (methods); It is already stated in Line 119-120 (introduction).
ANSWER
It has been deleted
COMMENT
- Tables 1-3 can be moved to supplementary material.
ANSWER
A new Supplementary Material Section has been created
COMMENT
- Line 151-156 and Line 289-293: Same information is presented twice in the methods sections. This manuscript has several such issues, which make it very lengthy and difficult to follow. It needs substantial improvement.
ANSWER
The manuscript has been rewriten
COMMENT
- The manuscript lacks a discussion of the results.
ANSWER
A discussion section has been added
COMMENT
- Present only the key findings and implications of the study in the conclusion section.
ANSWER
The conclussion section has been corrected
The authors appreciate the exhaustive work done by the editor and the reviewers who have served to improve the article a lot.
Round 2
Reviewer 3 Report
I appreciate the authors' effort in improving the manuscript. However, after going through the revised manuscript, I am not convinced that it can be published because the major shortcomings in the previous version are still there in the revised manuscript (see my comments in the previous version). The manuscript structure and flow is poor, and I encourage the authors reframing and resubmitting it.